# Taking into Account both Explicit Conduits and the Unsaturated Zone in Karst Reservoir Hybrid Models: Impact on the Outlet Hydrograph

**Lucie Dal Soglio [1], Charles Danquigny [1,2,\*] , Naomi Mazzilli [1] , Christophe Emblanch [1] and Gérard Massonnat [2]**

[1]  Avignon Université, INRAE, UMR EMMAH, F-84000 Avignon, France; lucie.dalsoglio@gmail.com (L.D.S.); naomi.mazzilli@univ-avignon.fr (N.M.); christophe.emblanch@univ-avignon.fr (C.E.)

[2]  Total, CSTJF, Avenue Larribau, 64018 Pau CEDEX, France; gerard.massonnat@total.com

\*  Correspondence: Charles.Danquigny@Univ-Avignon.fr

**Abstract:** The main outlets of karst systems are springs, the hydrographs of which are largely affected by flow processes in the unsaturated zone. These processes differ between the epikarst and transmission zone on the one hand and the matrix and conduit on the other hand. However, numerical models rarely consider the unsaturated zone, let alone distinguishing its subsystems. Likewise, few models represent conduits through a second medium, and even fewer do this explicitly with discrete features. This paper focuses on the interest of hybrid models that take into account both unsaturated subsystems and discrete conduits to simulate the reservoir-scale response, especially the outlet hydrograph. In a synthetic karst aquifer model, we performed simulations for several parameter sets and showed the ability of hybrid models to simulate the overall response of complex karst aquifers. Varying parameters affect the pathway distribution and transit times, which results in a large variety of hydrograph shapes. We propose a classification of hydrographs and selected characteristics, which proves useful for analysing the results. The relationships between model parameters and hydrograph characteristics are not all linear; some of them have local extrema or threshold limits. The numerous simulations help to assess the sensitivity of hydrograph characteristics to the different parameters and, conversely, to identify the key parameters which can be manipulated to enhance the modelling of field cases.

**Keywords:** karst; hydrodynamics; unsaturated zone; vadose zone; hybrid model; numerical experiments; hydrograph

## 1. Introduction

Most near-surface carbonate karst systems host groundwater reservoirs that supply freshwater to 20–25% of the global population [1]. Deeper carbonate formations contain around 60% of the world's conventional petroleum [2]. Despite increasing pressure on resources stored in karst reservoirs and the consequent need for sustainable management tools, modelling fluid dynamics in karst systems continues to be a challenge.

Specific karst features, especially conduit networks, are difficult to consider explicitly in models. In addition to their high heterogeneity and anisotropy at all scales that they overprint to the medium, karst conduits may undermine the hypothesis of the Darcian flows that are classically assumed for underground flows. Additionally, the high level of contrast between the hydraulic properties of the different media combined with the size and continuity of karst features makes it difficult to identify a representative elementary volume (REV) for the characterization of properties and upscaling. Moreover,

the hierarchical organisation of the conduit network concentrate flows to one or a few outlets [1]. These outlets are usually river springs whose hydrographs (spring discharge versus time, Figure 1a) integrate all hydrologic processes occurring in the reservoir to varying degrees and with various delays. For instance, the characteristics of the conduits, their density and connectivity and the structure of the conduit network affect the system response and the hydrograph shape [3–7]. The distribution of the recharge between diffuse and concentrated flows and the exchanges between the matrix and conduits also control the characteristics of the hydrograph, such as peak discharge and base flow [8,9]. This integrative role of hydrographs combined with good accessibility make springs favourable monitoring points. Consequently, spring hydrographs constitute primary variables to study karst systems [10–16] or calibrate numerical models [17–20]. Such discharge rate time series nevertheless differ from the usual piezometric monitoring approaches used to constrain groundwater models.

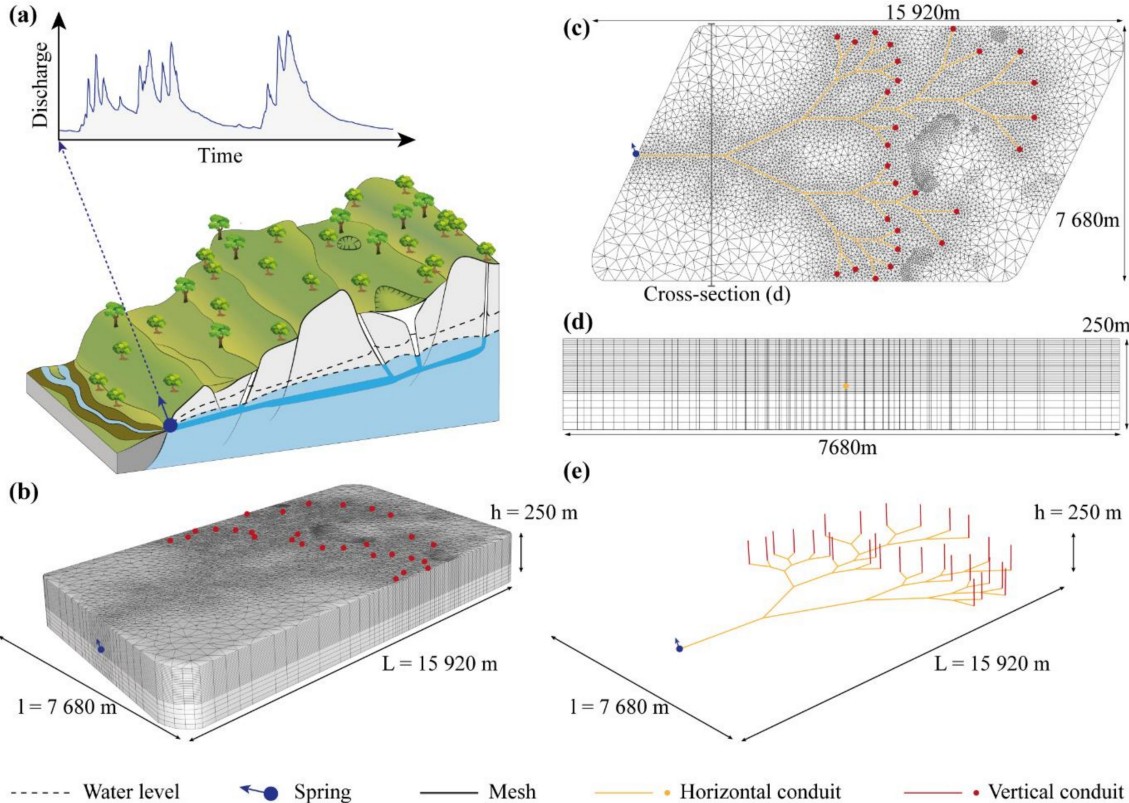

**Figure 1.** Schematic representation and corresponding model of karst system: (**a**) illustrations of flows towards a spring and hydrograph; (**b**) mesh of the model with locations of the vertical conduits (red points) and the outlet (blue arrow); (**c**) horizontal slice of the mesh with projection of the conduits and location of cross-section (**d**); (**d**) vertical cross-section of the mesh with location of the terminal conduit; (**e**) karst conduits network of the model.

The importance of the vertical structuration of karst on flow properties and processes at the reservoir scale is widely acknowledged [1,21–24]. Notably, the epikarst and transmission zone constitute very different subsystems, whose petrophysical properties differ enough to be distinguished in the models [25]. The epikarst is the near-surface weathered zone of the karst system [1]. Its porosity may reach 10%, while its hydraulic conductivity is generally higher than $10^{-5}$ m·s$^{-1}$ and tends to be isotropic due to alteration processes [26,27]. The transmission zone constitutes the relatively unaltered part of the unsaturated zone, where water mainly flows vertically towards the saturated zone. In the transmission and saturated zones, at the scale of the flow unit, the matrix porosity and hydraulic conductivity are usually less than 2% and $10^{-4}$ m·s$^{-1}$ respectively [24]. Flow processes in the unsaturated zone (soil, epikarst and transmission zone) can vary greatly in time and space [28–31]. Variable connectivity

inside the flow path network controls the infiltration processes [19,29,32–34]. Flows in the unsaturated zone can be either direct through conduits or delayed because they slowly circulate in the matrix [32]. The karst unsaturated zone may therefore act as a main storage reservoir [35,36], whose complex functioning largely affects the shape of hydrographs [26,36–43]. However, the unsaturated zone is rarely represented explicitly in models of karst hydrodynamics [17,44–49]. Most modelling studies only consider the saturated zone of the aquifer [18,20,50–55].

Introducing all these karst specificities into numerical models is difficult. Considering only physically-based 3D models, to date, aquifer-scale karst hydrodynamics have mostly been modelled using equivalent porous medium approaches [18,50,56]. These modelling methods represent the entire karst aquifer (matrix, fractures and karst conduits) as a single equivalent porous medium in which only Darcy's law applies. This simplification corrupts the simulated global response [5]. The relevance of such models is therefore dependent of the scale of the problem studied and that of reservoir heterogeneity [18,50,57]. In an opposite way, other modelling techniques enable the explicit representation of discrete channel networks. They allow the simulation of turbulent flow in karst conduit networks with complex geometry while neglecting the storage and flows in the matrix. These models are thus mostly dedicated to fractured reservoirs or conduit flow-dominated karst systems [58]. Taking into account both a mature karst conduit network and highly capacitive matrix requires a dual media approach [52,59]. In double continuum models, matrix and karst conduits are considered as two equivalent porous media linked by exchange terms. Such a dual representation does not solve all the difficulties as, in most cases, karst conduits are represented through an equivalent porous medium with Darcy flow. Moreover, the exchange term between the matrix and conduits cannot be measured and may be difficult to calibrate [47].

Hybrid models have arisen recently; by coupling a 3D equivalent porous medium representation of the matrix on a grid with networks of discrete 2D fractures or 1D conduits, they hold promise for a realistic representation of karst geometries [20,25,51,55,60,61]. They allow the separate and explicit consideration of some large conductive discontinuities that upscaling rules make it difficult to encompass in the equivalent porous medium representation [62]. Some hybrid models allow different flow physics in karst conduits to be taken into account [61,63–65]. However, in another paper [25], we reported the difficulty of considering both turbulent flows in the conduits and unsaturated flows in the matrix. We nevertheless showed the ability of hybrid models to simulate karst hydrodynamics in unsaturated conditions and to reproduce most processes that occur at the conduit scale and that are reported in the literature correctly. Moreover, we highlighted how varying the model parameters affects the flow processes and the exchanges between the matrix and conduits in both the epikarst and the transmission zone. Hybrid models thus seem mature enough, and their availability through market software makes them easy to apply [66,67].

Therefore, the question arises of whether approaches taking into account both unsaturated subsystems and explicit karst conduits enhance the simulation of both hydrodynamics at the karst reservoir scale and the hydrograph at the outlet. This paper focuses on the impact of such a configuration and the related parameters on the reservoir scale response, especially the spring hydrograph. First, this approach requires the capacity to distinguish different behaviours in the hydrograph shape, and particularly to determine the key descriptors of this response. Then, we study how these descriptors vary as functions of model parameters, particularly regarding the range of responses that we can expect from models whose parameters are consistent with literature values and how each subsystem, epikarst or transmission zone affects the model response. Based on modelling methods, results and commonly accepted concepts from the literature [25], we build a 3D hybrid model of a hypothetical karst aquifer; we assess and compare the hydrographs resulting from the simulation of recharge events for various sets of parameters. This work seeks to provide modellers with a range of parameters, guidelines and useful tips to enhance the modelling of field cases.

## 2. Materials and Methods

The next sections present the 3D hybrid model built to study the response of a hypothetical karst reservoir, including an unsaturated zone and conduits, to a recharge event. First, we present the hypothetical model and highlight the main characteristics of the considered karst system. Secondly, we focus on flow equations and parameters. Finally, the evaluation criteria of the several simulations are presented.

*2.1. Description of the Hybrid Model and the Considered Karst Specificities*

The model represents a hypothetical carbonate aquifer, as illustrated in Figure 1a, with a network of karst conduits following a branchwork pattern from the top of the model to the outlet [68]. A set of vertical conduits crossing the unsaturated zone drain towards a single outlet through a network of horizontal conduits in the saturated zone represents this network (Figure 1e). The system has a catchment area of 100 km$^2$ and a uniform thickness (250 m for the reference model). The outlet elevation is at 120 m. The Figure 1 presents several views of the model.

This finite elements model was built with FEFLOW 7.0 by DHI WASY (https://www.mikepoweredbydhi.com/products/feflow) [66,69]. It is composed of 30 layers; i.e., 31 slices with 6585 vertices per slice (Figure 1c). The slice spacing is 10 m in the unsaturated zone and 20 m in the saturated zone (Figure 1d). The mesh is refined around all discrete features to ensure convergence (Figure 1b,c); the cell size ranges from 10$^{-3}$ km$^2$ to 10$^{-1}$ km$^2$. The mesh cells support the porous fractured matrix, while a selection of mesh edges supports the discrete features that represent large conductive karst conduits. Both are homogeneous.

A uniform recharge flux is applied on the top of the model, whereas the discharge is controlled by a Dirichlet boundary condition equal to 150 m at the outlet of the conduit network. Other external faces are no-flow boundaries.

For the sake of comparison, several features are the same as those in [25]: the recharge flux, the structure of the unsaturated zone with conduits crossing it vertically, the flow equations and the model parameters excepted for the conduits. All these characteristics of the model are presented below.

*2.2. Flow Equations and Model Parameters*

In the unsaturated zone, we applied the Richards equation [70] to simulate the variably saturated water flow in the model's matrix:

$$\frac{\partial}{\partial x}\left[K(\psi)_{xx}\frac{\partial h}{\partial x}\right] + \frac{\partial}{\partial y}\left[K(\psi)_{yy}\frac{\partial h}{\partial y}\right] + \frac{\partial}{\partial z}\left[K(\psi)_{zz}\frac{\partial h}{\partial z}\right] \pm U = \frac{\partial \theta}{\partial t}, \tag{1}$$

where t is time (s), x, y and z are the spatial coordinates (m) (positive upwards), $\theta$ is the volumetric water content (-), h is the hydraulic head (m), $K(\psi)$ is the unsaturated hydraulic conductivity (m·s$^{-1}$) in the function of the pressure head ($\psi$) and U is the sink-source term (s$^{-1}$).

Our approach requires the definition of constitutive relationships for saturation as well as the relative permeability. However, the huge heterogeneity of fractured and karstified carbonate rocks causes not only petrophysical heterogeneity but also complex variations of capillary forces and saturation over short distances, making it difficult to assess these relationships at the mesh-cell scale. This is a poorly addressed issue in the literature and remains a challenge [25,70,71]. Based on the literature, we applied the Van Genuchten model with constant and uniform parameters (Table 1). The water content is equal to

$$\theta(\psi) = \theta_r + \frac{\theta_s - \theta_r}{\left[1 + (\alpha\psi)^n\right]^m}, \tag{2}$$

where $\theta_r$ and $\theta_s$ are residual and saturated water contents (-), respectively, and $\alpha$ (cm$^{-1}$), n and m are empirical parameters. The moisture content equals porosity multiplied by saturation. The relative hydraulic conductivity $K_r$ (-) in the unsaturated zone follows this relation:

$$K_r = S_e^{0.5}\left[1 - \left(1 - S_e^{\frac{1}{m}}\right)^m\right]^2.$$
(3)

where $S_e$ is effective saturation, generally defined as [72]

$$S_e = \frac{\theta - \theta_r}{\theta_s - \theta r}.$$
(4)

The hydrograph changes depending on whether laminar or turbulent flow is considered in the conduits [64]. However, several tests showed that the simulation fails if we consider both Richards and Manning–Strickler equations for the matrix and the conduits, respectively. Therefore, we applied the Darcy law to simulate the flow in the conduits that are always fully conductive. Thus, the product of the cross-section area by hydraulic conductivity, the so-called flow capacity, is the key parameter for advection in conduits. Moreover, matrix–conduit exchanges are implicit. Indeed, such kinds of finite elements hybrid models compute hydraulic heads on mesh nodes that define both cells supporting the equivalent porous medium and edges supporting the discrete features.

For the purpose of this study, we varied several parameters of the model, one at a time, around a reference simulation (Table 1): the thickness (Thk), porosity ($\Phi$) and hydraulic conductivity (K) of the epikarst (EK: Thk$_{EK}$, $\Phi_{EK}$, K$_{EK}$) and transmission zone (TZ), respectively, and the flow capacity of the conduits (KS). Petrophysical values for the saturated zone (SZ) are assumed to be equal to those of the transmission zone ($\Phi_{TZ-SZ}$, K$_{TZ-SZ}$). Due to the variable flooding of the epiphreatic zone, the thicknesses of the transmission zone and the saturated zone vary while their sum remains constant. The boundary condition at the outlet constrains the initial thickness of the saturated zone. Thus, the initial thickness of the transmission zone (Thk$_{TZ}$) is the only geometrical parameter of interest for the lower subsystems. According to the literature, hydraulic conductivity is isotropic only in the epikarst. In the other subsystems, the ratio between the horizontal hydraulic conductivity (K$_{TZ-SZ}$) and the vertical hydraulic conductivity is equal to 10, as usually assumed.

The discrete features of the model represent the major conduits whose size and flow capacity increase with the scale of the model. Flow capacity is also set to preserve the interest in considering explicit discrete conduits by avoiding too conductive discrete features that would be equivalent to fixed-head boundary conditions [5,25] while preserving the contrast of conductivity between the matrix and conduits. To respect this compromise, several tests led us to consider a reference value for flow capacity equal to 100 m$^3$·s$^{-1}$. For comparison, this value is 1000 times greater than that of the conduit scale model of Dal Soglio et al. [25], while the area of the model is 100 times greater. Nevertheless, the preliminary results led us to retain only values larger than this reference among all the tested values. Indeed, the simulated groundwater level locally exceeds the ground level if considering smaller values for conduit flow capacity together with other reference parameters. This highlights the importance of this parameter and the difficulty of calibrating it.

Table 1 presents the reference values and the range of variations for all the variable parameters of the model, based on a literature overview [25]. It must be emphasized that most of the values found in the literature are not related to a given measurement volume and are generally independent of the support. The upscaling issue is generally not addressed, and this should be kept in mind when interpreting the simulation results.

**Table 1.** Value ranges for the properties of karst systems and karst modelling reported in the literature and model parameters [25].

| Subsystem | Property (Units) | Values and Ranges of Values [1] from Literature | Model's Values and Range of Values Min–Ref–Max |
|---|---|---|---|
| Epikarst (EK) | Thickness $Thk_{EK}$ (m) | (0; >30) [24]<br>(few meters; $10^{-15}$) [23]<br>(3; 10) [1]<br>(8; 12) [73] | 0–20–35 |
| | Porosity $\phi_{EK}$ (-) | (0.05; 0.1) [24,74]<br>(0.1; 0.3) [27]<br>>0.2 [1] | 0.01–0.1–0.25 |
| | Horizontal [2] hydraulic conductivity $K_{EK}$ (m·s$^{-1}$) | $(10^{-7}; 10^{-4})$ [41]<br>$10^{-5}$ [39]<br>$(5 \times 10^{-5}; 10^{-3})$ [75]<br>$(2 \times 10^{-4}; 2 \times 10^{-3})$ [76]<br>$10^{-3}$ [77]<br>>1000 * $K_{TZ-SZ}$ [78] | $10^{-5}$–$10^{-2}$–$10^{-1}$ |
| Transmission and saturated zones (TZ–SZ) | Thickness $Thk_{TZ}$ (m) | depending on the field site, usually tens of meters,<br><20; <50 [77]<br>up to 700 [32] | 30–80–130 |
| | Porosity $\phi_{TZ-SZ}$ (-) | (0.004; 0.01) [1]<br>0.005 [79]<br>(0.01; 0.02) [80]<br>(0.024; 0.3) [81] | 0.005–0.01–0.025 |
| | Horizontal [2] hydraulic conductivity $K_{TZ-SZ}$ (m·s$^{-1}$) | $(10^{-10}; 7 \times 10^{-5})$ [81]<br>$(10^{-7}$ [46]; $10^{-6}$ [1,47]) [75]<br>$(5 \times 10^{-7}; 5 \times 10^{-6}$ [39]) [9]<br>$(10^{-6}$ [1,47]; $10^{-4}$ [79]) [80]<br>$(10^{-5}; 10^{3})$ [17] | $10^{-7}$–$10^{-5}$–$10^{-3}$ |
| Conduit (C) | Diameter D (m) | (0.08; 15) [64]<br>(2; 10) [60] | Flow Capacity $A_C$ * $K_C$ (m$^3$·s$^{-1}$) $10^{-2}$–$10^{-1}$–$10^{1}$ |
| | Section $A_C$ (m$^2$) | (<1; >100) [82] | |
| | Hydraulic conductivity $K_C$ (m·s$^{-1}$) | $(6 \times 10^{-5}; 4 \times 10^{-1})$ [81]<br>$(10^{-1}; 10)$ [17,75]<br>(3; 10) [80]<br>10 [9,47] | |
| Van Genuchten Model | Coefficient $\alpha$ (m$^{-1}$) | $(3.28 \times 10^{-3}; 6.23 \times 10^{-1})$ [44]<br>$3.65 \times 10^{-2}$ [47,49]<br>$10^{-2}$ [17,46] | $3.65 \times 10^{-2}$ |
| | Empirical parameter n (-) | (0.01; 3) [44]<br>1.83 [47,49]<br>2 [17,46] | 1.83 |
| | Residual water content $\theta_r$ (-) or Residual water saturation $S_r$ (-) | $\theta_r = S_r = 0$ [46]<br>$\theta_r \in (0.01; 0.05)$ [44]<br>$S_r = 0.05$ [47]<br>$\theta_r = 0.171$ [17] | $S_r = 0.05$ |

[1] Ranges of values from the literature are shown in parentheses. [2] When anisotropy is considered, values concordant to the strata are presented.

## 2.3. Simulations and Evaluation Criteria

For all the parameters sets, a single recharge event is simulated. The recharge boundary condition is a uniform flux (i.e., without a focused recharge point on discrete features) applied on the top of the model. Initials conditions result from a steady simulation with a recharge equal to 0.5 mm/day. A single recharge event is added to this steady recharge at the beginning of the transient simulation, uniformly providing 100 mm in two days, represented by an isosceles triangle reaching a peak of 100 mm/day in one day. After numerous tests, this rather high event appeared to be the best compromise to illustrate the results of this study. Note that such intense rainfall events are commonly observed in the Mediterranean climate [83,84]. The simulation results reflect the hydrodynamic behaviour subsequent to this single recharge event.

The hydrograph resulting from a single precipitation event consists of a rising limb, a flood peak and a falling limb (Figure 2). In some cases, several peaks were seen to appear due to highly contrasted flows in the medium. Numerous parameters allow the description (e.g., peak flow value and time,

flow duration, recession coefficient and form factors as N-order moments) or analysis (e.g., memory effect, cut-off frequency) of the shape of this response. Thus, time-continuous approaches as a derivative of the discharge help us to deepen this analysis [8]. Figure 2 illustrates some of these parameters on a theoretical hydrograph.

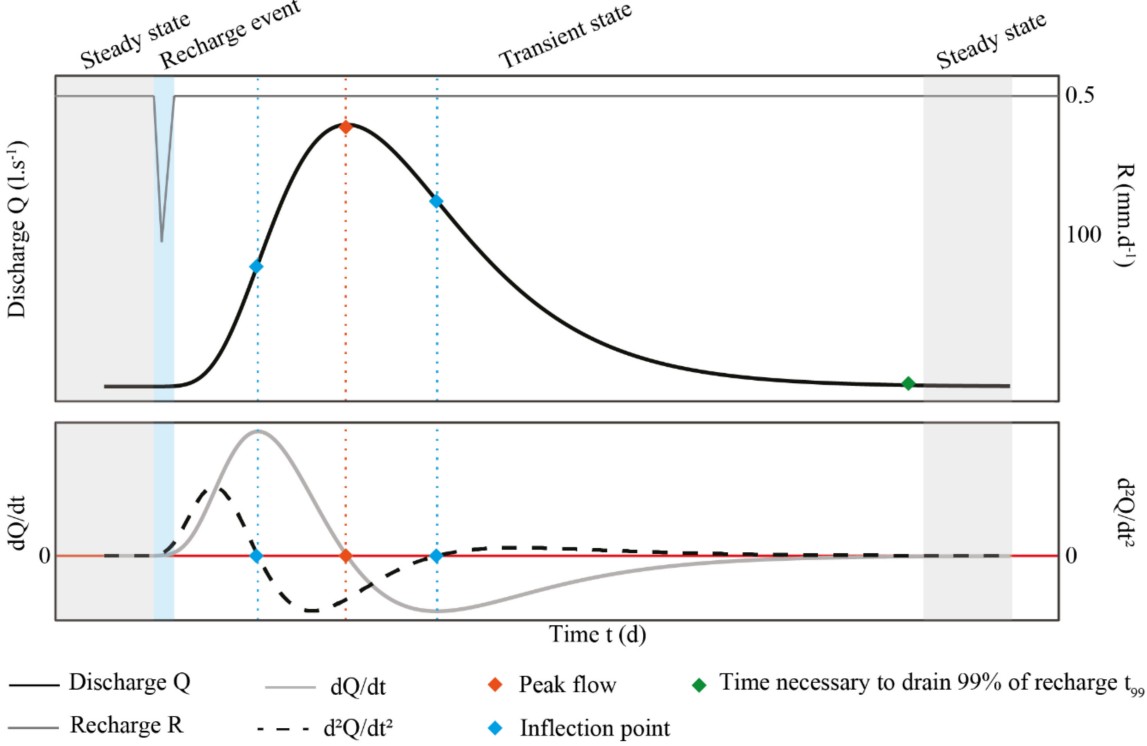

**Figure 2.** Theoretical hydrograph resulting from a single precipitation event.

In addition, skewness (third-order standardized moment) and kurtosis (fourth-order standardized moment) may help us to characterize the shape of the hydrographs. For a discrete discharge time series, the n-order standardized moment is calculated as follows:

$$x = \frac{1}{m}\Sigma_{t=1}^{m}\left(\frac{Q_t - \mu}{\sigma}\right)^n,$$ (5)

where Q is the discharge, m is the number of time steps, $\mu$ is the mean and $\sigma$ is the standard deviation of the discharge time series.

Here, we evaluated the model's response regarding the following hydrograph characteristics (Figure 2):

- Peak flow (maximum discharge value); in some cases, several local extrema are identified;
- Time after the event until peak flow;
- Discharge duration;
- Third order moment (skewness), which describes the shape of distribution;
- Fourth order moment (kurtosis), which is a flattening coefficient.

All moments and statistics are calculated from the beginning of the recharge event to $t_{99}$, the time necessary to drain 99% of recharge event water to the spring. Several tests have concluded that water drained after $t_{99}$ does not affect the results.

As they have been studied widely and for a long time, some karst systems have become usual examples to illustrate the diversity of karst systems and their responses. In the literature, Torcal, Aliou, Baget, Fontestorbes and Fontaine de Vaucluse systems [1] provide hydrograph characteristics

that are used as references to assess the realism of the numerical experiments presented in this paper. In addition to other parameters calculated by Marsaud [85], skewness and kurtosis, have been calculated for selected unit hydrographs to evaluate the parameter ranges for each system (Figure 3). Unsurprisingly, skewness values are positive because the discharge distribution is shifted towards the left. Skewness values range between 0.3 and 1.8. Kurtosis data extend from 2 to 5, reflecting the spread of the hydrograph response. Detailed information about Torcal in Spain [16,86], Aliou, Baget, and Fontestorbes in the French Pyrenees [15,87–90] and Fontaine de Vaucluse in southeastern France [36,91–94] is available in the literature.

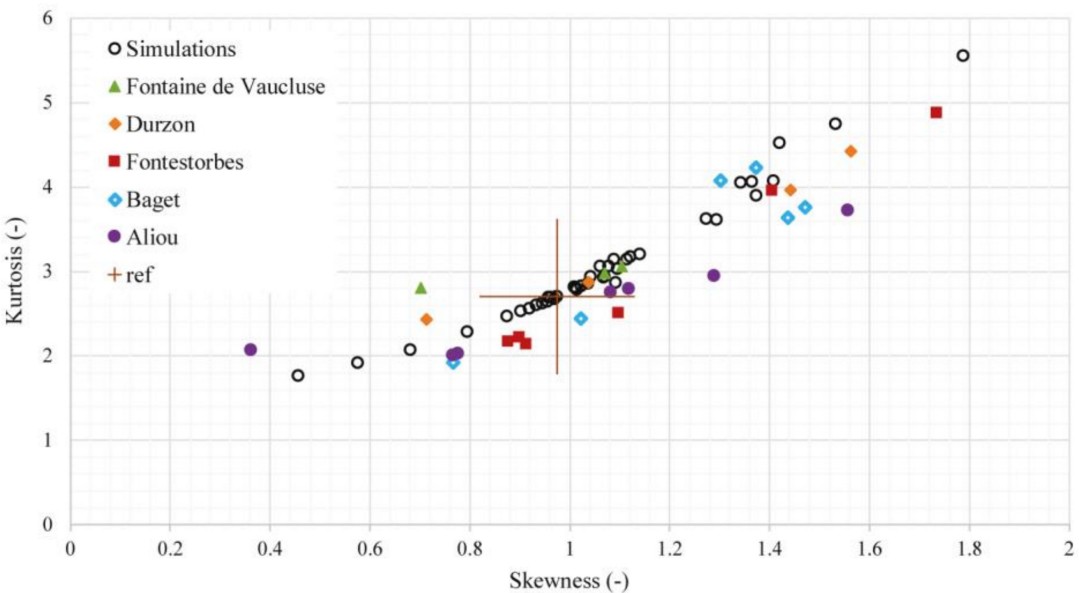

**Figure 3.** Moments calculated for several karst systems and the numerical simulations.

## 3. Results and Discussion

### 3.1. Overview of Simulation Results and Hydrograph Typology

Our simulations provide hydrographs of various shapes with one or two peaks that are more or less embedded. Hydrographs with two distinct peaks highlight a bimodal transit time distribution—i.e., a clear, early and balanced separation of recharge between quick and diffuse flows—likely between the conduits and matrix. In the other cases, either one kind of flow is preponderant or the heterogeneity drives the flows through various pathways, spreading the distribution of transit times. Building on simulation results and those from the literature, we define a classification with five different hydrograph shapes (Figure 4) to facilitate the analysis of results. Type 1 (Figure 4a) corresponds to preponderant diffuse circulation in the continuum, whereas Type 5 (Figure 4e) corresponds to water circulating predominantly in the karst network. Both have only one discharge peak. When the matrix flow competes with conduit flow, three intermediate types can be distinguished: type 2, with one peak preceded by an inflection (Figure 4b); the bimodal type 3, with two discharge peaks (Figure 4c); and type 4, with one peak followed by an inflection (Figure 4d). Distinguishing the hydrograph types requires the identification of possibly small inflections. This might be difficult without the support of information provided by the first and second derivatives of discharge as a function of time [8].

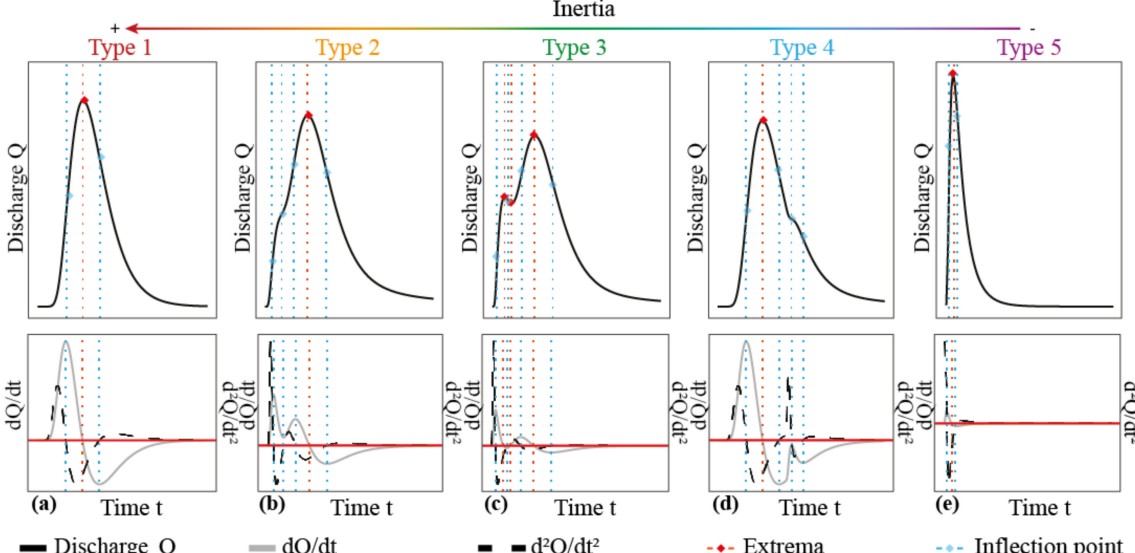

**Figure 4.** Hydrograph typology established on the basis of the simulation results and the literature.

Regarding representative karst systems from the literature [1,16,85,89], aquifers with a low degree of karstification, such as the Torcal system (Spain), can be likened to the type 1 systems presented here. Aquifers with a high degree of karstification, such as the Aliou system (France), correspond to type 5. Between these two extreme end-members, the response of the Baget system corresponds to type 4, and that of Fontestorbes is type 2 or type 3. For more detail, see the work presented by Marsaud [85] which characterized the hydrographs of these systems.

Figures 5–8 show the simulation results for different varying parameters. Figure 5 focuses on conduit flow capacity, while Figures 6–8 are dedicated to matrix parameters. Figures 5a and 6 highlight the hydrograph diversity that a single model can produce by varying only one parameter at a time. The scatter plots of Figures 5–8 present the selected hydrograph characteristics as functions of the different varying parameters. All plots (hydrographs and related characteristics) are coloured according to the hydrograph typology: red for type 1, orange for type 2, green for type 3 and purple for type 5. There is no occurrence of a type 4 hydrograph in the paper.

Note that the reference simulation is visible on all the plots and results in a Type 2 hydrograph (Figure 6). For comparison with other simulation results, 99% of the single recharge event drains within 2160 days, with a peak discharge equal to 745 L·s$^{-1}$ at the 68th day. The values of the quantities obtained in the reference model should not be considered in absolute terms but only by comparison with the simulations carried out for other parameter sets. Effectively, these values depend both on the structure of the hypothetical aquifer constructed for the simulation and on the values retained to quantify all the parameters of the model. The long duration of draining observed in the simulation can be also related to the flow processes in the unsaturated matrix under variably saturated conditions. Indeed, matrix flows last longer in variably saturated conditions. As drainage occurs, saturation and hydraulic conductivity decrease, slowing the flow accordingly. Moreover, conduits draining the surrounding medium may dry it locally and create less conductive zones around them. This thereby limits the area of influence of conduits in the unsaturated zone. In the hypothetical aquifer matrix, the distance to the nearest conduit is highly variable, with some areas being very distant from the karst network, notably in each corner of the model (Figure 1c), which reinforces such behaviours. Recharge that is not drained towards the karst network flows vertically through the transmission zone, which acts as a buffer zone spreading the temporal distribution of the recharge event. For instance, in a model representing only the vicinity of a vertical conduit, with the same vertical organization of the medium and comparable properties, the two-day recharge event at the top of the model spans several dozens of days at the bottom of the transmission zone [25]. This result highlights the importance of the

karst network structure and the distribution of distances from matrix to the nearest conduits in the model response.

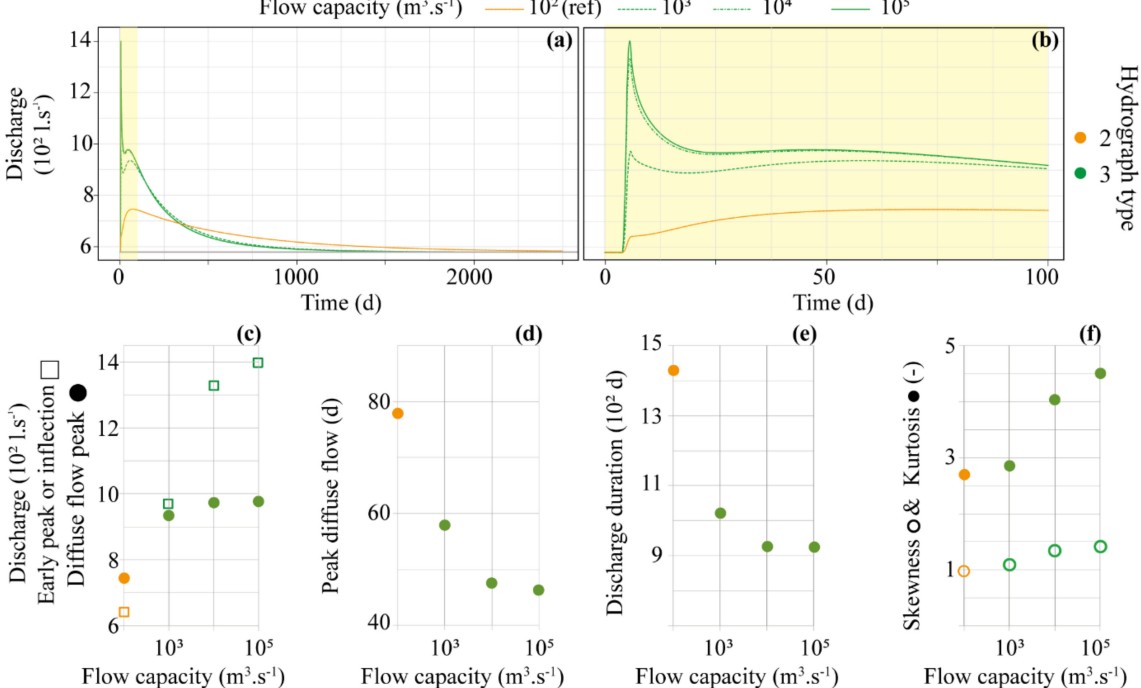

**Figure 5.** Hydrograph characteristics for various conduit properties. Plots are coloured according to the hydrograph typology: orange for type 2 and green for type 3.

As in the reference simulation, most hydrographs are type 2: discharge increases rapidly to an inflection point, after which the increase is smaller than in the first phase and spreads out over time. The first inflection reflects a relatively small fast-flow component that adds to a broad distribution of transit times and pathways of diffuse flows, which is to a certain extent related to the structure of the hypothetical aquifer.

Type 3 hydrographs have two distinguishable peaks: the first early peak is representative of more significant fast flows than for type 2 hydrographs, while the secondary peak indicates a narrow distribution of transit times corresponding to diffuse flows. Type 3 occurs for instances of a high flow capacity in the conduits (Figure 5), low porosity in the epikarst (Figures 6a and 7) or high porosity in the transmission zone (Figure 8). The first two configurations concentrate flow towards nearby conduits and thus limit pathway spreading and transit times [25]. A high porosity in the transmission zone limits its saturation by recharge events. Therefore, the transmission zone is relatively less conductive, which also promotes flow concentration towards conduits in the upper zone.

Type 1 hydrographs occur when a highly effective conductivity of the transmission zone is favoured; i.e., for instances with high conductivity (Figure 6c) but also low porosity (Figure 8) or low thickness in this zone (Figure 6b). Conversely, type 5 hydrographs occur only for very low values of hydraulic conductivity in transmission and saturated zones (Figure 6c).

Compared to the simulation results, actual systems produce more complex hydrographs that reflect the complexity of the flow network architecture of the different media and the variability of the recharge conditions. Moreover, varying only one parameter at a time limits the range of responses simulated, as exemplified by the majority of hydrographs being of the same type to that of the reference simulation. However, these results highlight the importance of slope variations in hydrographs, linked with the recharge occurrence and repartition between matrix or conduit-dominated flows [8]. The set of simulations screening different parameters contributes to the identification of the flow processes and subsystem characteristics that cause either behaviour.

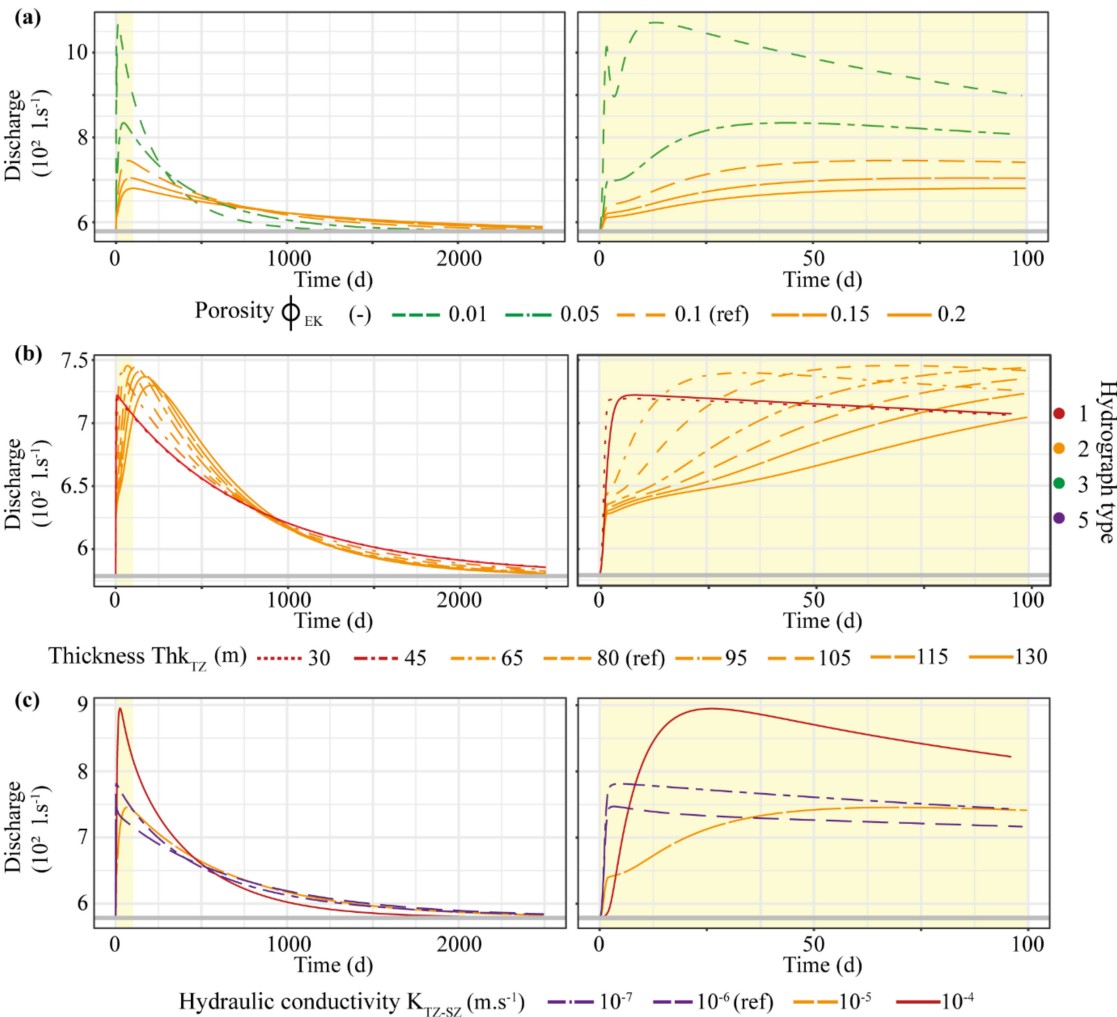

**Figure 6.** Hydrograph results for various parameter sets. (**a**) Hydrographs for varied porosities in the epikarst $\phi_{EK}$; (**b**) hydrographs for varied thicknesses of the transmission zone $Thk_{TZ}$; (**c**) hydrographs for varied hydraulic conductivities of the transmission and saturated zones $K_{TZ\text{-}SZ}$. Plots are coloured according to the hydrograph typology: red for type 1, orange for type 2, green for type 3 and purple for type 5.

### 3.1.1. The Role of Epikarst Parameters

The Figure 7 shows the hydrograph characteristics obtained after having tested different values for several parameters of the epikarst subsystem. This confirms previous results: decreasing storage capacity by decreasing porosity or thickness or, to a lesser extent, by increasing hydraulic conductivity heightens the flow concentration towards conduits and the fast flow component [25]. Above all, it shows the consequences of local processes on hydrographs. Without an epikarst—i.e., for an epikarst thickness of 0—the early peak is the lowest. When the epikarst is explicitly present, low porosity or low thickness promotes drainage towards conduits, low storage and a short transit time with narrow distribution. These behaviours produce more asymmetric hydrographs (i.e., with higher skewness). The hydraulic conductivity of the epikarst primarily affects the overall discharge duration with a threshold for higher values. The higher the conductivity, the larger the quantity of water drained towards conduits and the higher the discharge rate of the early peak. Increasing the hydraulic conductivity also tends to reduce the contrast between the matrix and conduit properties, which produces more spread and less asymmetric hydrographs. Thus, kurtosis and skewness decrease as a function of the epikarst's hydraulic conductivity.

The parameter variation ranges cover the usual values from the literature, but they are also relatively small and of the same order of magnitude as the typical measurement uncertainties. However, hydrograph characteristics do not vary linearly with the epikarst parameters (Figure 7). For a thickness equal to 15 m, the peak flow time reaches a maximum while skewness and kurtosis reach minima. Several thresholds can be observed; for instance, skewness and kurtosis reach a threshold value for the highest values of the three parameters. The lowest values, equal to 1 and 2.8 for skewness and kurtosis, respectively, are reached for porosity above 0.10, hydraulic conductivity above $10^{-2}$ m·s$^{-1}$ and thickness equal to 10 m. Finally, in this configuration, the epikarst parameters that have the greatest individual effect on the hydrograph are porosity and thickness for values ranging between [0.01; 0.10] and [0; 10 m], respectively. However, varying several parameters at a time should produce combined effects that could eventually be more important.

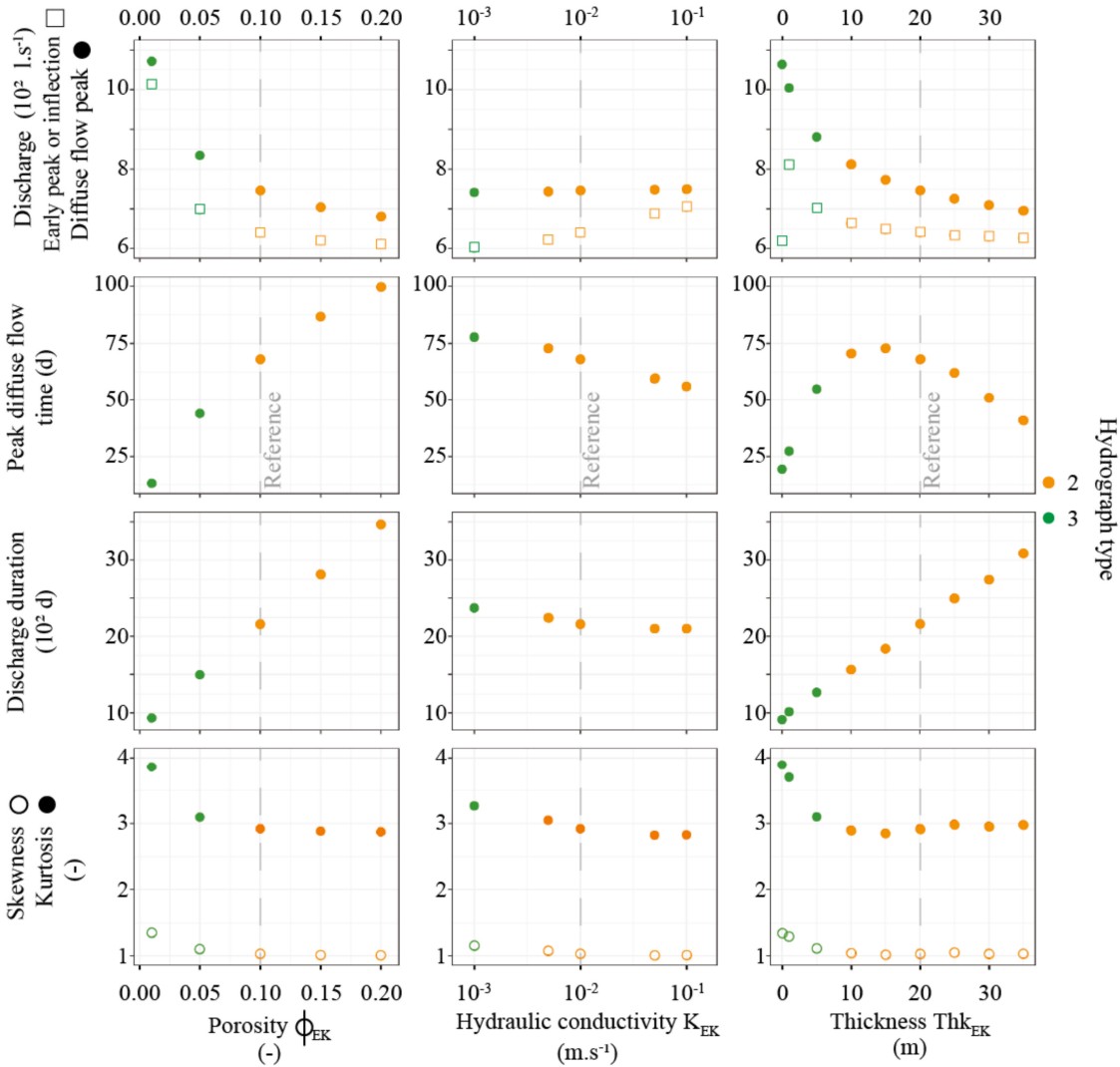

**Figure 7.** Hydrograph characteristics for various values of several properties in epikarst. Plots are coloured according to the hydrograph typology: orange for type 2 and green for type 3.

### 3.1.2. The Role of Transmission and Saturated Zones Parameters

Figure 8 plots the characteristics of hydrographs after having evaluated different values for several parameters of the transmission zone and the saturated zone. As with the epikarst, the ranges of the variation of parameters cover the usual values from the literature. Despite these relatively small ranges, the responses are very different for the resulting hydrographs from type 1 to type 5 (e.g., Figure 6c).

As water flows preferentially through the most permeable zones, the elevated hydraulic conductivity in the transmission zone promotes vertical drainage through the continuum and limits drainage towards conduits in the epikarst [25]. This twofold effect induces a major variation of hydrograph features as a function of the hydraulic conductivity of transmission and saturated zones. Varying the conductivity over three orders of magnitude is enough to obtain the extreme types of hydrographs. Indeed, among the tested sets of parameters, type 5 occurs only with a very low conductivity (below $10^{-6}$ m·s$^{-1}$) of the transmission and saturated zones.

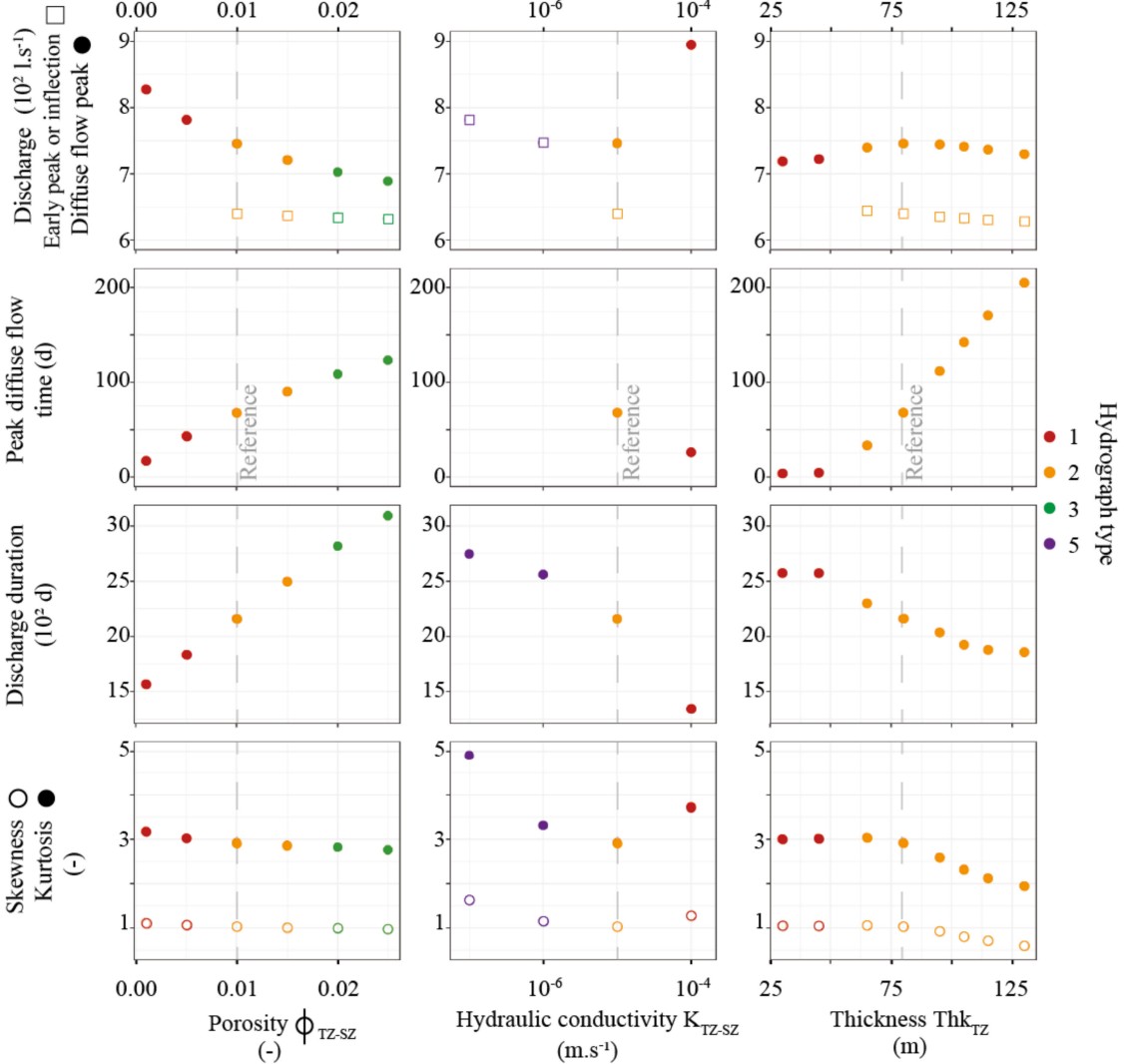

**Figure 8.** Hydrograph characteristics for various values of several properties in the transmission zone and the saturated zone. Plots are coloured according to the hydrograph typology: red for type 1, orange for type 2, green for type 3 and purple for type 5.

Porosity and thickness are key factors in storage capacity. Increasing the capacity should result in higher inertia, lower peaks and a longer discharge duration. These relationships are verified and almost linear for porosity. The thickness also affects transit times and therefore flow repartition. The plots of hydrographs characteristics as a function of the thickness of the transmission zone show local extrema and thresholds with changes of hydrograph type. Indeed, for small thicknesses of the transmission zone (here, below 50 m), the hydrograph characteristics are almost constant. The hydrographs are type 1 with only one visible peak and a long tail highlighting a broad distribution of transit times and pathways of diffuse flows. Increasing the thickness of the transmission zone makes the conduit

competitive as the highway for long journeys. The thickness affects the water distribution between conduits, and the matrix as identified with conduit scale models. Consequently, for higher thicknesses (here above 50 m), hydrographs are type 2 with an early flow distinct from the variably delayed distribution of the diffuse flows. The time of the diffuse flow peak increases while the discharge duration decreases towards a threshold value with an increased thickness of the transmission zone (Figures 6b and 8). Early flow is distinguishable from a porosity or thickness greater than 0.01 or 65 m, respectively. However, these parameters have little effect on the early flow characteristics.

Finally, the resulting hydrographs reflect the two functions of the transmission zone; i.e., a possible horizontal barrier at the interface with the epikarst and a vertical pathway competing with the vertical conduit [25].

## 4. Evaluation of Models

The numerical experiments presented in this paper aim to assess the interest and quantify the impact of explicit representations of both karst conduits and unsaturated zones in karst reservoir modelling. We built a single hypothetical model whose geometry and parameters were chosen with the condition of being consistent with the literature. The simulations performed cover a wide range of behaviours, which allows us to highlight the major contributions and limitations of this modelling approach.

### 4.1. Model Assumptions

Hybrid models are able to reproduce many characteristics of the karst aquifer structure. However, as with any modelling approaches, hybrid flow modelling relies on assumptions and simplifications, which provide a compromise between realism, the ability to provide input data and computational tractability. For example, the conductive discrete features represented in hybrid models are only a small fraction of the actual karst network. Indeed, only the most important drains or an upscaled representation of the preferential flow network can be considered in models because of limitations in both knowledge of the system and numerical capabilities.

In this study, we considered homogeneous recharge and homogeneous hydrodynamic properties for both media, which both minimize preferential pathways and flow hierarchy. Most authors choose an a priori repartition of recharge between the matrix and conduit network to favour concentrated flow [9,14]. Here, the flow concentration towards the conduits is enabled by the epikarst subsystem [25]. Contrasting behaviours are obtained by varying the epikarst flow properties. The effects of the topography and dip are not considered here, although they may play a major role at the reservoir scale in recharge distribution and the concentration of flow towards conduits.

Turbulent flow is characteristic of karst conduits and can be accounted for by using the Manning–Stickler equation [95,96]. However, the importance of taking turbulence into account varies with the size and roughness of the simulated conduits; thereby, applying laminar flow equations is sufficient for saturated, mature karst systems with well-developed conduit networks [65]. In unsaturated flow conditions, recent work successfully coupled variably saturated flow modelling in a matrix with turbulent flow modelling in the conduit [61]; the scale investigated was nevertheless smaller than in the present case. Here, preliminary tests revealed the difficulty of coupling the Richards equation in the equivalent porous medium and the Manning–Strickler equation in the discrete features. We therefore used Darcy law to simulate conduit flow. Conduits are assumed to be fully conductive whatever their saturation state, which seems to be consistent with the expected properties of the mainly vertical karst conduits in the vadose zone, which never reach saturation.

Only one formula with only one set of parameters was tested regarding the constitutive relationship between the saturation and the relative permeability of the matrix. The thorough assessment of this latter relation would deserve dedicated studies, including datasets of measurements on rock samples, relationship fitting with data and upscaling rules considering small-scale heterogeneity as fractures or vugs. Likewise, assessing the value of the conduit flow capacity is difficult. It is bounded by the

concerns of (i) establishing a conductivity contrast between the matrix and conduits, (ii) ensuring sufficient drainage of the recharge for the lower bound, and (iii) avoiding the creation of an overly conductive conduit that would be efficiently replaced by fixed-head boundary conditions for the higher bound. Above all, this parameter must be consistent with the object or the processes it represents.

*4.2. Scaling Issues*

Providing realistic values for model parameters is a concern when dealing with scaling issues. Upscaling, which should be a key issue in such systems, is surprisingly often neglected when property values are proposed. Laboratory measurements are generally performed for rock samples whose volume is smaller than the representative elementary volume (REV), if it exists, and whose selection criterion is mainly based on the homogeneity of the sample, leading to the avoidance of specific carbonate features such as fractures, vugs or fossils [97]. At the larger scale, the equivalent permeability value for a given larger volume strongly depends on the geometric organization of the permeability field within this volume, which often lacks characterization [98]. Moreover, considering hybrid models requires thresholds in hybrid implicit–explicit representations of fractures and karst features to be partitioned [62]: smaller drains should be lumped with the rock matrix in the upscaling process to limit the number of discrete features explicitly represented in the model. Finally, dealing with variably saturated flow modelling may raise the most topical scaling issues, with both theoretical [99] and methodological [100] unanswered questions.

In this work, parameter values were chosen in a usually admitted range based on the literature review, assuming that the values in the literature—which are generally independent of the support and not actually measured—are effectively representative of the volumes to be quantified for the model grids.

*4.3. Evaluation of Models Outputs*

4.3.1. The Need of Hydrographs Descriptors

The effect of varying parameters has been quantified on the simulated hydrographs. In order to assess the differences between the hydrographs resulting from the various simulations, we defined some characteristics of interest: the peak flow, time after the event until peak flow, discharge duration, skewness and kurtosis. Moreover, we proposed a hydrograph classification based on inflections points and—more generally—slope changes.

Only four of the five proposed types of hydrographs were obtained with the model. As type 3 and type 5 occur, the absence of the intermediate type 4, which includes an early peak followed by an inflection point and corresponds to a common observed shape of hydrographs, is probably related to the need for a delicate parametrization to produce it, but may also highlight some flaws in the model setup. For instance, a matrix area distant from the karst network would have poor drainage due to the use of uniform parameters, with the consequence of giving an important weight to the diffuse flow component and the possible over-sensitivity of the related parameters, which should therefore be finely controlled to produce a type 4 inflection point. This simplification also contributes to explaining the high number of type 2 hydrographs including a wide distribution of the diffuse flow component. These considerations highlight the impact on the hydrograph shapes of large-scale heterogeneity in the karst conduit distribution.

4.3.2. Matching Model Outputs with Field Measurements

Even if the modelled aquifer is hypothetical, the resulting hydrograph characteristics seem to be realistic in terms of some aspects for an aquifer with a catchment area of 100 km$^2$ and a uniform thickness of 250 m: the peak flow value varies between 597 and 1063 L·s$^{-1}$, the peak flow time varies between 4 and 204 days and the discharge duration varies between 912 and 3464 days. We use skewness and kurtosis descriptors for the shape of the hydrographs. Figure 3 shows kurtosis as a function of

skewness for all the simulations and for hydrographs from the well-known karst systems described in Section 4.3.2. The values from simulations are consistent with the values from field sites. They cover the same ranges, and the reference simulation is almost centred. The long discharge durations could possibly be questioned, but these can probably be related to the huge uncertainty related to the upscaling issue. This is likely accentuated here by the model structure, with a poorly karstified area far from the represented karst network. These results nevertheless highlight the important delaying effect of the unsaturated zone.

## 5. Conclusions

This work focuses on the consideration of several karst zones and explicit conduits in the reservoir modelling of a karst aquifer at a large scale. Together with the saturated zone, the models include the unsaturated zone, in which a distinction is made between the epikarst and the transmission zone. More generally, the paper addresses the issue of performing realistic simulations of flows in complex media such as a karst. Based on numerous flow simulations on a hypothetical karst aquifer model, we investigated the ability of hybrid models to simulate spring hydrographs that are usual observations in karst studies. Moreover, we explored the relationships between model parameters and the relevant hydrograph characteristics.

In addition to classical characteristics such as the maximum discharge value and corresponding time, we have considered other key features, such as inflections, but also the overall hydrograph shapes through parameters such as skewness and kurtosis or the proposed classification. All these features are definitely useful for both the study of hydrographs and the analysis of flow simulation results.

At the reservoir scale, the hydrograph incorporates the hydrodynamics of the entire system and therefore constitutes a primary output to assess or calibrate a model. Varying parameters affect pathways distribution and transit times to various extents, which results in a large variety of hydrograph shapes. The relationships between model parameters and hydrograph characteristics are not all linear: some of them have local extrema (e.g., peak flow time vs thickness of epikarst) or threshold limits (e.g., all characteristics vs thickness of the transmission zone). The numerous simulations help to assess the sensitivity of hydrograph characteristics to the different parameters. For instance, the discharge duration is more sensitive to the storage capacity (porosity and thickness) of the epikarst than to its conductivity. More generally, the storage capacity appears to be at least as important a feature as hydraulic conductivity in flow distribution. Therefore, this study should help researchers involved in modelling to identify the key parameters to modify to reproduce observations from actual sites.

Finally, the hybrid models are able not only to reproduce flow processes at the interface between the matrix and conduit [25] but also to simulate the overall response of complex karst aquifers. Several avenues for improvement nevertheless arise, in particular with regard to the problems of flow physics up-scaling in both unsaturated porous media and conduits.

**Author Contributions:** Conceptualization, L.D.S., C.D. and N.M.; methodology, L.D.S., C.D. and N.M.; software, L.D.S.; validation, L.D.S., C.D. and N.M.; formal analysis, L.D.S.; investigation, L.D.S.; writing—original draft preparation, L.D.S.; writing—review and editing, L.D.S, C.D., N.M., G.M.; visualization, L.D.S.; supervision, C.D., N.M., G.M., C.E.; project administration, C.D., G.M.; funding acquisition, C.D., G.M. All authors have read and agreed to the published version of the manuscript.

**Funding:** This study was supported by Total S.E.

**Acknowledgments:** The authors would like to thank Total S.E. for its support and for permission to publish this paper. This work also benefited from fruitful discussions within the Karst observatory network (SNO KARST) initiative of the INSU/CNRS, which seeks to strengthen knowledge sharing and promote cross-disciplinary research into karst systems at the national scale.

**Conflicts of Interest:** The authors declare no conflict of interest.

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
