# Peer review of "Taking into Account both Explicit Conduits and the Unsaturated Zone in Karst Reservoir Hybrid Models: Impact on the Outlet Hydrograph"

_water, doi:10.3390/w12113221_

Round 1

Reviewer 1 Report

This is a potentially valuable manuscript dealing with an interested topic. The approach is correct, even if temporal perched aquifers in the unsaturated zone are not addressed.

The topic is well known to some readers, but for others some definitions to bound term should be included, as the epikarst; this term has different meanings.

The manuscript has many mentions to the paper Dal Soglio [31]. which is a previous step to the present manuscript, but still unpublished. The essential from this paper have to be given al the beginning and the main results that will considered later. Repeated citation is to be avoided

A more carefully detailed description of the simulated media is needed to assure the reader grasp what is explained as results.

In figure 1 (129) figure 1 is too obscure

(207) Check the definition of the N-order moment and define variables more carefully

(256) the caption of this figure and others tells the color code, but no all colors appear or there are doubts if the are really present, specially red and orange. Note this in the caption

263 variables parameters....please explain

280-281 unclear sentence

282 the distance of matrix from conduits should be clarify referred to to the mesh or some comparative value

354 clarify for a better understanding

364 result in higher    add in

369 clarify the sentence

Author Response

Dear Reviewer,

Please find attached our reponse to your review.

Best regards,

Charles Danquigny,

on the behalf of the authors.

Reviewer 2 Report

This paper presents a solid piece of work. However, some elements of the paper should be revised to do the work justice. More specific comments are in an attached file

Author Response

(The authors gave the same response as above.)

Round 2

Reviewer 1 Report

Most previous comments have been addressed. The paper is now more clear and to the point. A few comment are included in the annotated manuscript, which can be managed by the editor together with the authors

Author Response

Dear reviewer,

We thank you for your last detailed proofreading which allowed us to correct the last mistakes of the article. We have corrected them all.

Best regards,

The authors.